# Limited protection against early-life lung murine cytomegalovirus infection results from deficiency of cytotoxic CD8 T cells

Luís Fonseca Brito[1,2,3], Eleonore Ostermann[3], Silvia Tödter[1], Sanamjeet Virdi[4], Daniela Indenbirken[4], Renke Brixel[3], Ramon Arens[5], Adam Grundhoff[4], Wolfram Brune[3], Felix Rolf Stahl[1,2,3]*

1 Institute of Immunology, University Medical Center Hamburg-Eppendorf, Hamburg, Germany, 2 Hamburg Center for Translational Immunology, University Medical Center Hamburg-Eppendorf, Hamburg, Germany, 3 Department of Virus-Host Interaction, Leibniz Institute of Virology, Hamburg, Germany, 4 Research Unit Virus Genomics, Leibniz Institute of Virology, Hamburg, Germany, 5 Department of Immunology, Leiden University Medical Center, Leiden, The Netherlands

* f.stahl@uke.de

## Abstract

Differential antiviral T cell immunity in early life impacts the clinical outcome of Cytomegalovirus (CMV) infection, but the underlying mechanisms are not well understood. T cells are known to be involved in protection from CMV disease. To further elucidate these mechanisms, we used a model of respiratory murine CMV (MCMV) infection and adoptive T cell transfers to characterize MCMV-specific T cell responses in early life. We analyzed the effector T cell differentiation using single-cell RNA sequencing and assessed the local pulmonary cytokine milieu. We found delayed enrichment of early-life murine MCMV-specific CD8 T cells due to a general deficiency of αβ T cells. Adoptive transfer of naïve adult T cells into neonates did not protect from lung MCMV infection due to generation of non-cytotoxic CD8 effector T cells. Furthermore, key cytokines required for effective CD8 T cell priming were absent in early life. Supplementation with these cytokines enhanced infection control by transferred adult T cells. The effector function of adult-primed T cells was not disrupted in neonates. Together, this study suggests defective CD8 T cell priming in neonates as a factor explaining the higher risk for MCMV lung disease in the early-life phase.

## Author summary

Cytomegalovirus infection causes a significant health burden in neonates and therapeutic options are limited. Preclinical models are important to study the underlying mechanisms of this viral disease. In the present study, we found that activation of T cells, a critical component of the antiviral immune response, is impaired in the respiratory tract of neonatal mice. T cells integrate several

**Data availability statement:** Single-cell RNA-seq data has been deposited at ENA (accession PRJEB70246) and code used for generation of the figures has been uploaded on https://github.com/LIV-NGS/T_cell_CMV. All other data are provided within the manuscript and the supporting information files.

**Funding:** This work was supported by the Deutsche Forschungsgemeinschaft (https://dfg.de) SFB1713 project number 534829736 (FRS and WB), STA 1549/1-2 (FRS), and BR 1730/9-1 (WB); the Werner Otto Stiftung (https://www.werner-otto-stiftung.de) (FRS); and the Medical Faculty of the University of Hamburg (https://www.uke.de) (FRS). We acknowledge financial support from the Open Access Publication Fund of UKE - Universitätsklinikum Hamburg-Eppendorf. The funders had no role in study design, data collection and analysis, decision to publish, or preparation of the manuscript.

**Competing interests:** The authors have declared that no competing interests exist.

extracellular signals to initiate an effector response and some of these are absent in lungs of MCMV-infected neonatal mice. In contrast, effector T cells obtained from adult hosts are protective in neonates. Our data indicate that alterations in the early phase of T cell activation interfere with neonatal antiviral immunity.

## Introduction

Early-life Cytomegalovirus (CMV) infection, in humans and mice, is characterized by prolonged viral replication and increased risk of disease [1–3]. While CMV is generally considered to be an opportunistic pathogen causing predominantly mild or asymptomatic infection in immunocompetent hosts [4], congenital CMV or infection of preterm low-birth-weight newborns is the most important infectious cause of permanent disabilities in infants [5]. To date, there is no protective vaccine available, and therapeutic options are limited, not curative, and bear the risk of severe adverse effects [6]. Accordingly, congenital CMV causes a significant health burden [7] and urgently requires investigation of disease pathophysiology.

CMV induces exceptionally strong T cell responses that are crucial for virus control and regularly lead to ~15% of the circulating CD8 T cell pool being specific for CMV epitopes [8]. Thus, it is reasonable to assume that age-related differences in anti-CMV T cell immunity are involved in the increased vulnerability in early life. Indeed, neonates exhibit an altered composition of peripheral blood T cell compartments and differential antigen-recognition patterns compared to adults [9]. However, virus-specific effector T cells are present in newborns after infection with human CMV (HCMV) [10] and in neonatal mice challenged with murine CMV (MCMV) [11,12], indicating that, in general, the very young host can acquire anti-CMV T cell immunity. Nevertheless, the presence of these effector T cells does not preclude CMV disease, and it is currently not known to what extent these early-life effector T cells exhibit protective functions.

Application of MCMV into the respiratory tract of neonatal mice leads to a characteristic immune response with influx of leukocytes into the lung and formation of nodular inflammatory foci [13,14]. Neonatal nodular inflammatory foci consist of myeloid cells, including antigen-presenting cells, which can prime CD8 T cells [13,15]. However, adoptive transfer of naïve antigen-specific CD8 T cells into MCMV-infected neonates does not protect from lung infection. This contrasts with MCMV infection of T cell-deficient immunocompromised adult mice, where adoptive transfer of naïve polyclonal T cells reduces virus loads in lungs [16]. The underlying mechanisms of the differential T cell effects *in vivo* remain to be defined.

Here, we delineate age-related differences in anti-MCMV T cell immunity. We demonstrate that the expansion of endogenous MCMV-specific T cells early in life is delayed due to an overall low frequency of peripheral T cells. Adoptive transfers of naïve adult T cells into neonates generated CD8 effector T cells that were low in cytotoxicity, and this correlated with their non-protective function in the young host. These non-cytotoxic CD8 T cells resulted from altered early-life T cell priming conditions.

The impaired T cell response was not observed after adoptive transfer of effector T cells obtained from MCMV-infected adults, indicating that early life antiviral immunity was impaired at the level of T cell priming rather than the T cell effector phase. This study indicates that the early-life susceptibility to MCMV lung infection is linked to dampened cytotoxic T lymphocyte (CTL) induction.

## Results

### Delayed expansion of MCMV-specific CD8 T cells in early life

To investigate the pathophysiology of the high early-life risk for CMV disease, we infected neonatal and adult mice by administering MCMV to the respiratory tract [13]. We utilized an MCMV recombinant expressing mCherry and Gaussia luciferase that has previously been used in several in *vivo* studies to monitor and quantify MCMV infection [16–19]. Hypothesizing that T cell immunity is involved in the delayed MCMV control in the lungs of neonates (Fig 1A) [13], we tracked MCMV-specific T cells in peripheral blood using pMHC-I tetramer staining. We detected CD8 T cells recognizing MCMV immunodominant epitopes M38, M45, and m139 [20] after seven days post-infection (dpi) in adults (Fig 1B and 1C), consistent with previous reports [11,21,22]. In neonates, MCMV-specific CD8 T cells could not be detected before 10 dpi, and the frequency of these cells was lower (Fig 1B and 1C). This observation was in line with a previous study where we had observed a delayed expansion of M45-specific CD8 T cells in neonatal mice [13]. We also analyzed the accumulation of M25-specific CD4 T cells [23] and detected more cells stained with an MHC-II tetramer in infected than in control animals, both in adults and neonates (S1A and S1B Fig). To follow up on the low numbers of MCMV-specific CD8 T cells in neonates, we assessed the presence of lymphocytes in the spleen of non-infected animals during the very early postnatal life period. On the day of birth, we found populations of B220$^+$ B and CD3$^-$NK1.1$^+$ natural killer (NK) cells, but very few CD3$^+$ T cells (Figs 1D, 1E, and S1C). Within the first days of life, the microanatomy of the spleen underwent a distinct organization, and the numbers of CD3$^+$ T cells expressing TCRβ and CD4 or CD8β in the white pulp increased (Fig 1D and 1E). The relative number of B cells also increased, albeit to a lesser extent than T cells, whereas the frequency of NK cells remained rather stable within the first four weeks of life. By calculating the absolute number of lymphocytes relative to the animal body weight we found an approximately 10.000-fold increase of T cells in the spleen within the first week of life (Fig 1F). Taken together, we reasoned that the low precursor frequency of T cells in the early life could explain the delayed enrichment of MCMV-specific CD8 T cells and reduced control of lung infection.

### Naïve adult T cells adoptively transferred into MCMV-infected neonates do not protect from infection

To test whether low numbers of T cells accounted for the delayed control of MCMV infection in early life, we adoptively transferred purified polyclonal CD3$^+$ T cells obtained from secondary lymphoid organs of adult act-eGFP mice into neonates (Fig 2A). In a previous study, we observed that adoptively transferred T cells migrated into neonatal lungs within 24 hours [13]. Transferred cells homed to lymphoid organs and led to an approximately 60-fold increase of T cells in spleens within two days post intraperitoneal injection, reaching similar cell per body weight numbers as found in four-week-old mice (Fig 2B and 2C). Next, we combined adoptive T cell transfers with MCMV infection and found the adult T cells to acquire CD44 membrane expression and proliferate in neonatal mice in response to virus challenge (Figs 2D, 2E, and S2A). Moreover, a fraction of the CD8 T cells was MCMV-specific, indicating that the increase of T cell precursors indeed augmented the numbers of MCMV-specific CD44$^+$ effector T cells at 8 dpi (Fig 2F and 2G). MCMV causes the formation of nodular inflammatory foci (NIF) in the lung, in which T cells recognize antigens and participate in the control of infection [13,15]. Accordingly, adoptively transferred T cells accumulated in neonatal NIFs (Fig 2H). However, adult T cells did not decrease the number of MCMV-infected cells in neonatal lung NIFs (S2B Fig), and even transfer of high numbers of T cells did not reduce the viral load in the lungs (Fig 2I). Transfer of purified CD44$^+$ T cells from adult non-infected mice also did not protect against MCMV (Fig 2J). To further increase the number of antigen-specific T cells in neonatal NIFs, we infected the animals with

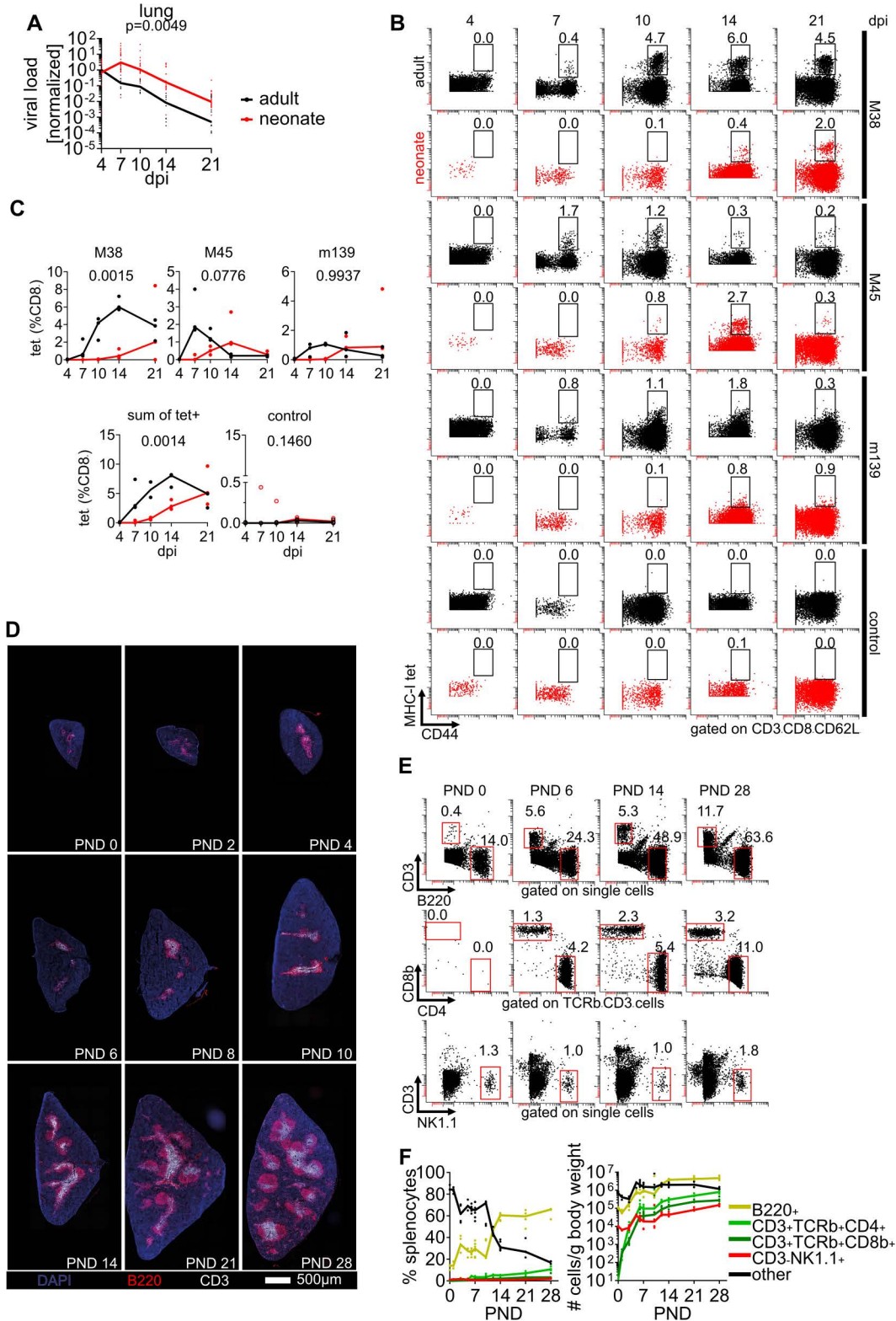

**Fig 1. Delayed enrichment of MCMV-specific CD8 T cells in early life. (A-C)** Neonatal and adult mice were infected with MCMV and analysed at indicated days post infection (dpi). **(A)** Lung viral loads of adult and neonatal mice after MCMV-infection. Data normalized to the median value at 4 dpi are displayed. **(B)** Representative flow cytometry plots and **(C)** pooled analysis of frequency of peripheral blood CD8 T cells stained for MCMV peptide

MHC-I tetramers as indicated. Non-infected mice were stained simultaneously with a pool of all three MHC-I tetramers as controls. **(D-F)** Non-infected mice were analysed for appearance of major lymphocyte populations. **(D)** Immunohistology of mouse spleens stained as indicated at post-natal days (PND) as indicated. **(E)** Representative flow cytometry plots of major lymphocyte populations in the spleen. **(F)** Frequency (left panel) and absolute number of cells normalized to animal body weight (right panel) of major lymphocyte populations in spleen. Data were acquired from more than two independent experiments (A, n=8-29 per time point; B and C, n=3 per time point; D-F, n=3-7 per time point). Lines in **(C)** and **(F)** connect the median value of each time point. Statistical difference between frequency of MCMV-specific CD8 T cells in **(C)** was calculated via a 2-way ANOVA, and the p value is provided above each graph.

MCMV-4DR, a recombinant virus encoding for epitopes recognized by OT-I and OT-II transgenic T cells, or with MCMV-2DR as a control (Fig 2K). MCMV-4DR infection of neonatal mice led to significantly higher proliferation of adoptively transferred naïve OT-I and OT-II cells than in mice infected with MCMV-2DR (Fig 2L and 2M). We found OT-I and, to a lower degree, OT-II cells in NIFs at 8 dpi (Fig 2N). As a control, we used adult *Rag2-/-Il2rg-/-* mice, which lack major lymphocyte populations. These mice were infected with either of the two MCMV recombinants and received the same number of naïve OT-I and OT-II cells as neonates. Importantly, this equals a 20-fold lower number of adoptively transferred T cells into adults, calculated per gram body weight. While antigen-specific T cells reduced viral loads in MCMV-4DR-infected *Rag2-/-Il2rg-/-* adult mice, there was no significant effect in neonatal mice or animals infected with MCMV-2DR (Figs 2O and S2C). Adoptive transfer of polyclonal T cells into adult *Rag2-/-* mice decreased viral loads when approximately 100 T cells were present in NIFs [16]. In neonates, although up to 250 OT-I T cells were found in NIFs, these cells failed to significantly reduce the number of infected cells or inflamed tissue (S2D – S2F Fig). The findings highlight that adoptive transfer of adult naïve T cells into neonates can compensate for the deficiency of this cell population in the early-life period, lead to antigen-specific proliferation, gain of CD44 expression, homing into sites of MCMV infection, but has no influence on virus load in the lung.

## Early-life MCMV infection induces a Th1-like CD4 T cell phenotype

Since transferred adult naïve T cells were unable to protect from early-life MCMV infection, we hypothesized that age-related factors modulate T cell differentiation and effector function. Therefore, we compared the phenotype of approximately 6000 adult T cells after priming in either adult or neonatal mice by single-cell transcriptome profiling (scRNA-seq). Adult GFP+ naïve T cells were adoptively transferred into MCMV-infected neonates. In our hands, recovery of adoptively transferred T cells into MCMV-infected adults was too low for reasonable analysis. Instead, endogenous adult T cells were used for comparison (Figs 3A and S3A). T cells isolated from the lungs of MCMV-infected adults and neonates exhibited a distinct transcriptome when compared to non-infected controls (Fig 3B). Expression of *Cd4* or *Cd8* was equally distributed across the four different conditions (S3A Fig) and positioned cells into two main clusters (Fig 3C). Most of the cells obtained from the MCMV-infected mice had low *Ccr7* expression indicating that they lost their naïve phenotype (Fig 3C). In line, we identified a higher frequency of T cells with shared T cell receptors (TCRs) after MCMV infection than in control animals, supporting the idea of antigen-dependent proliferation and emergence of effector T cells (Fig 3D). T cells isolated from uninfected neonatal and adult mice exhibited similar RNA expression profiles, indicating that under steady-state conditions, the phenotype of adult GFP+ cells in neonates is stable and closely resembles that of endogenous adult T cells (Fig 3E). In contrast, cells obtained from MCMV-infected mice exhibited a notable age-related difference in RNA expression (Fig 3E).

To gain insight into CD4 T cell differences *Cd4+Cd8a-* cell clusters were annotated according to their gene expression signatures into naïve (*Tcf7*, *Ccr7*, and *Sell*) and cycling (*Mki67* and *Pclaf*) T cells, regulatory T cells (Treg) (*Foxp3* and *Il2ra*), and T-helper 1 cells (Th1) (*Cd44*, *Ly6c2*, and *Cxcr6*) [24,25] (Fig 3F and 3G). All CD4 T cell subpopulations were found in the four different conditions (Fig 3H). MCMV infection decreased the relative number of naïve T cells, which, on the other hand, led to a higher frequency of cycling T cells and Th1 cells (Fig 3H). Tregs were found in similar numbers independently of age or infection (Figs 3H and S3B). However, there was a trend suggesting that adult T cells primed in MCMV-infected neonates were likely to differentiate into Th1 cells (Fig 3H), and these cells showed a distinct gene expression pattern when compared to Th1 cells found in adults (Fig 3I and S1 Table). There was higher expression of

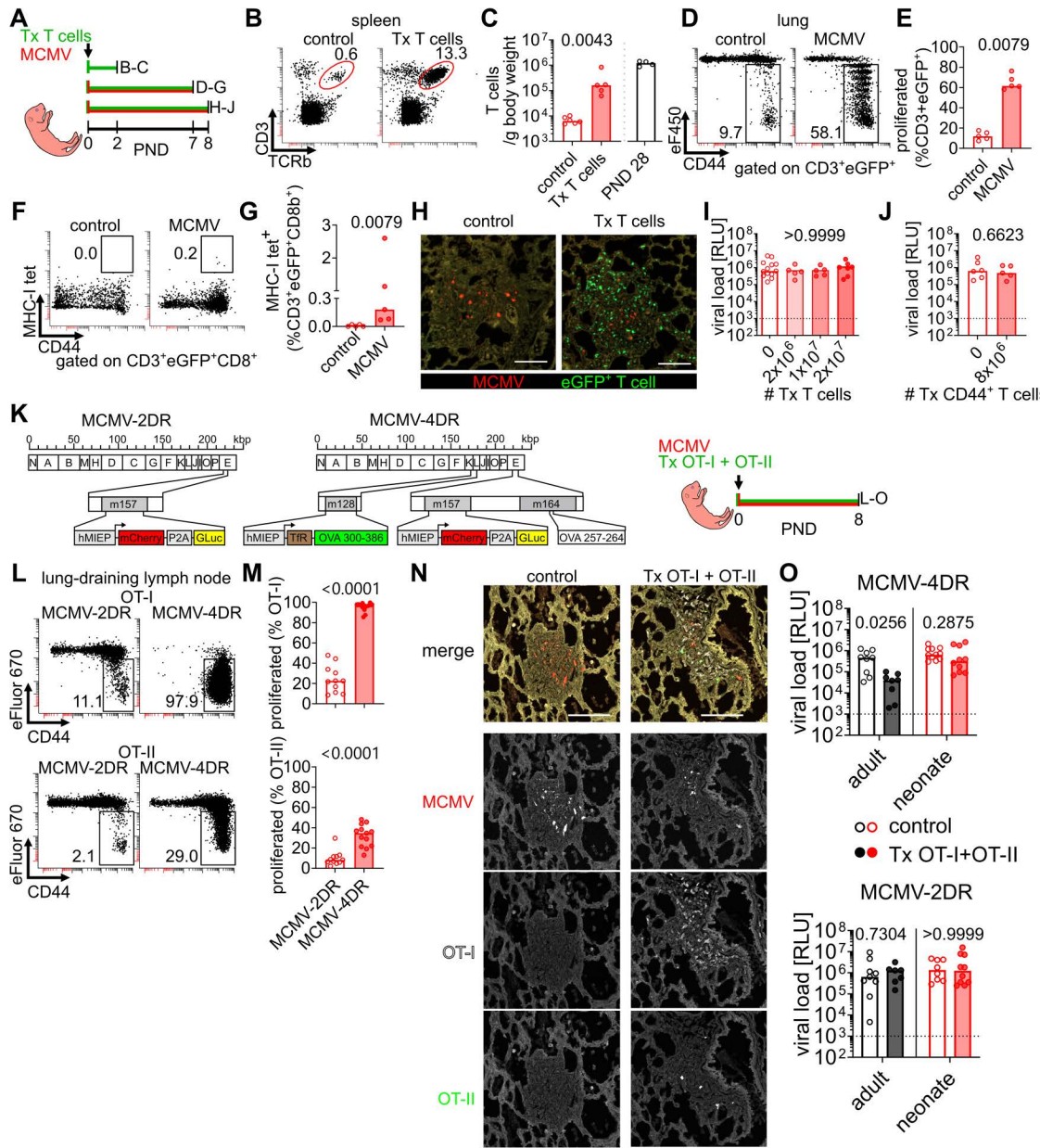

**Fig 2. Adoptive transfer of adult naïve T cells into neonates is not protective against MCMV. (A)** Experimental setup for **(B-J)**: **(B, C)** 2x10⁷ purified T cells obtained from adult eGFP transgenic mice (eGFP) were transferred (Tx) into neonatal mice and analysed after 2 days. **(D-G)** 10⁷ eGFP⁺ T cells were stained with cell proliferation dye eF450 and adoptively transferred into neonatal mice, which were simultaneously infected with MCMV. Control mice received cells but were not infected. Animals were analysed at 7 days after cell transfer. **(H,I)** Various numbers of eGFP⁺ T cells were adoptively transferred into neonatal mice, which were simultaneously infected with MCMV. Control mice received no T cells. **(J)** 8x10⁶ purified eGFP+CD44+T cells obtained from adult non-infected mice were adoptively transferred into neonatal mice, which were simultaneously infected with MCMV or buffer control. Animals were analysed at 8 days post infection (dpi). **(B)** Representative flow cytometry plots of spleen T cells. **(C)** Absolute numbers of T cells calculated to animal body weight 2 days after adoptive transfer. Cell numbers of non-treated PND 28 animals are depicted in Fig 1F for comparison. **(D)** Representative flow cytometry plots of eGFP⁺ T cells isolated from lungs, indicating eF450 signal loss with each proliferation cycle. **(E)** Quantitative analysis of proliferated eGFP⁺ T cells isolated from lungs. **(F)** Representative flow cytometry plots of eGFP⁺ CD8⁺ T cells isolated from lungs simultaneously stained with M38-, M45-, and m139-pMHC tetramers. **(G)** Quantitative analysis of MCMV-specific tetramer⁺ eGFP⁺ CD8⁺ T cells isolated from lungs. **(H)** Immunohistology of lungs indicating localisation of transferred eGFP+T cells in nodular inflammatory foci (NIF). **(I)** Lung viral loads after adoptive transfer of various numbers of naïve T cells. RLU, relative light units. **(J)** Lung viral loads after adoptive transfer of CD44+T cells from adult non-infected mice. **(K)** Experimental setup for **(L-O)**: MCMV-2DR contains no additional inserted sequences besides the fluorophore and luciferase reporters,

whereas MCMV-4DR encodes for sequences of chicken ovalbumin. Neonatal wildtype or adult Rag2-/-Il2rg-/- mice were infected (104 or 2x105 PFU, respectively) with either MCMV-2DR or MCMV-4DR and received adoptive transfers of 5x10$^6$ OT-I$_{eCFP}$ and 5x10$^6$ OT-II$_{eGFP}$ proliferation dye-labelled cells; control animals did not receive T cells. Animals were analysed at 8 dpi. **(L)** Representative flow cytometry plots of T cells isolated from lung draining lymph nodes indicating eF450 signal loss with each proliferation cycle. **(M)** Quantitative analysis of proliferated T cells. **(N)** Immunohistology of MCMV-4DR -infected neonatal lungs, indicating localisation of transferred OT-I$_{eCFP}$ and OT-II$_{eGFP}$ T cells next to MCMV-infected cells in NIF. **(O)** Lung viral loads after infection with MCMV-4DR (upper panel) or MCMV-2DR (lower panel) and adoptive transfer of OT-I and OT-II cells. Data were acquired from two or more experiments (B and C, n = 5-6; D-G, n = 5; H and I, n = 5-13, J, n = 5-6 L-O, n = 7-11 animals per group). Scale bars, 100 μm. Statistical differences between groups were calculated with Mann-Whitney U (C, E, G, L, and N) or Kruskal-Wallis **(I)** tests. The p values are provided above each graph. GLuc, *Gaussia* luciferase; hMIEP, human major immediate early promotor; OVA, chicken ovalbumin; TfR, transferrin receptor sequence.

pro-inflammatory, anti-apoptotic, and cell cycle-related genes in neonates (e.g., *Ifng, Birc5, Pclaf*) (Fig 3I). Moreover, CD4 T cells obtained from neonates had a higher cytotoxicity module score in the MCMV-infected groups (Fig 3J), but not in the controls (S3C Fig). Priming in MCMV-infected neonates led to a stronger clonal T cell response than in adults (Fig 3K), but with comparable relative distribution into the four different CD4 subpopulations (Fig 3L). Taken together, there were age-related differences in CD4 T cell priming with a stronger Th1 response in cells obtained from neonates.

### Early-life T cell priming impedes differentiation into cytotoxic CD8 T cells

Next, we annotated the *Cd8$^+$Cd4$^-$* CD8 T cell clusters into naïve (*Sell, Tcf7, Ccr7*), cycling (*Mki67, Pclaf*), and effector T cells (*Gzmb, Cd44, Ccl5*) [26,27] (Fig 4A and 4B). MCMV infection led to a decrease of naïve and an increase of cycling and effector CD8 T cells in both adults and neonates (Fig 4C). Comparison of effector T cells revealed differential gene expression with differentiation-promoting genes *Zeb2* and *Id2* and cytotoxicity-related genes such as *Ccl5, Gzmk*, and *Gzma* being more highly expressed in adults, whereas CD62L-encoding gene *Sell* and differentiation-repressing genes *Tcf7, Lef1,* and *Zeb1* were more abundant in neonates (Fig 4D and S2 Table). Accordingly, T cells isolated from neonates and adults were differently distributed within the effector T cell cluster (Fig 4E). Thus, we performed a sub-cluster analysis of CD8 T effector cells and named the according subpopulations as T$_{eff}$ 1 (high in, e.g., *Cd44, Ly6c2, Gzmm* but also *Sell*), T$_{eff}$ 2 (high in, e.g., *Ifit1, Ifit3*, and *Gzmb*), T$_{eff}$ 3 (high in, e.g., *Cxcr6, Cxcr3*, and *Ccl5*) and T$_{eff}$ 4 (high in, e.g., *Ccl5, Gzma, Cx3cr1, Zeb2*, and *Tbx21*) (Fig 4F and 4G and S2 Table). All T$_{eff}$1–4 subpopulations were present in neonates and adults (Fig 4H). However, T cells isolated from non-infected neonates clustered more as T$_{eff}$1 and less as T$_{eff}$4. In response to MCMV infection, changes in T$_{eff}$1 and T$_{eff}$2 were comparable between neonates and adults but there were fewer cells in neonates that acquired a T$_{eff}$3 and T$_{eff}$4 phenotype (Fig 4H). Accordingly, we found a higher frequency of lung CD8 T effector cells to be positive for membrane protein staining of CXCR6 and intracellular CCL5 in MCMV-infected adults than in neonates (S4A – S4C Fig). Next, module score analysis revealed that in adults, T$_{eff}$2–4 but not T$_{eff}$1 exhibited high cytotoxicity function in response to MCMV infection, indicating that antiviral CTLs are found within clusters T$_{eff}$2–4 (Figs 4I and S4D). In contrast, neonatal T$_{eff}$2 - the CD8 effector subpopulation that strongly increased in response to neonatal MCMV (Fig 4H) - had a low cytotoxicity module score (Fig 4I). Moreover, differentiation into cytotoxic T$_{eff}$3 and T$_{eff}$4 was minor in neonates when compared to adults (Fig 4H), explaining the overall lower expression of cytotoxicity-related genes in CD8 effector T cells in neonates (Fig 4D). Furthermore, clonal expansion in response to MCMV as an indicator for antigen-specific CD8 T cell response was higher in adults (Fig 4J). In neonates, clonally expanded CD8 T cells remained in a cycling state, whereas in adults most CD8 T cells acquired an effector phenotype (Fig 4K and 4L). Taken together, there was a significant age-related difference in the CD8 T cell effector response with lower clonal enrichment and cytotoxicity in MCMV-infected neonates than in adults, resulting in significantly fewer antiviral CTLs in early life.

### Ineffective early-life CD8 T cell immunity is associated with alterations in cytokine signaling conditions

The effector phenotype of antiviral CD8 T cells is highly influenced by the inflammatory cytokine milieu within the infected tissue [28]. Thus, we quantified cytokines known to be involved in CD8 T cell priming in lungs in steady state or in

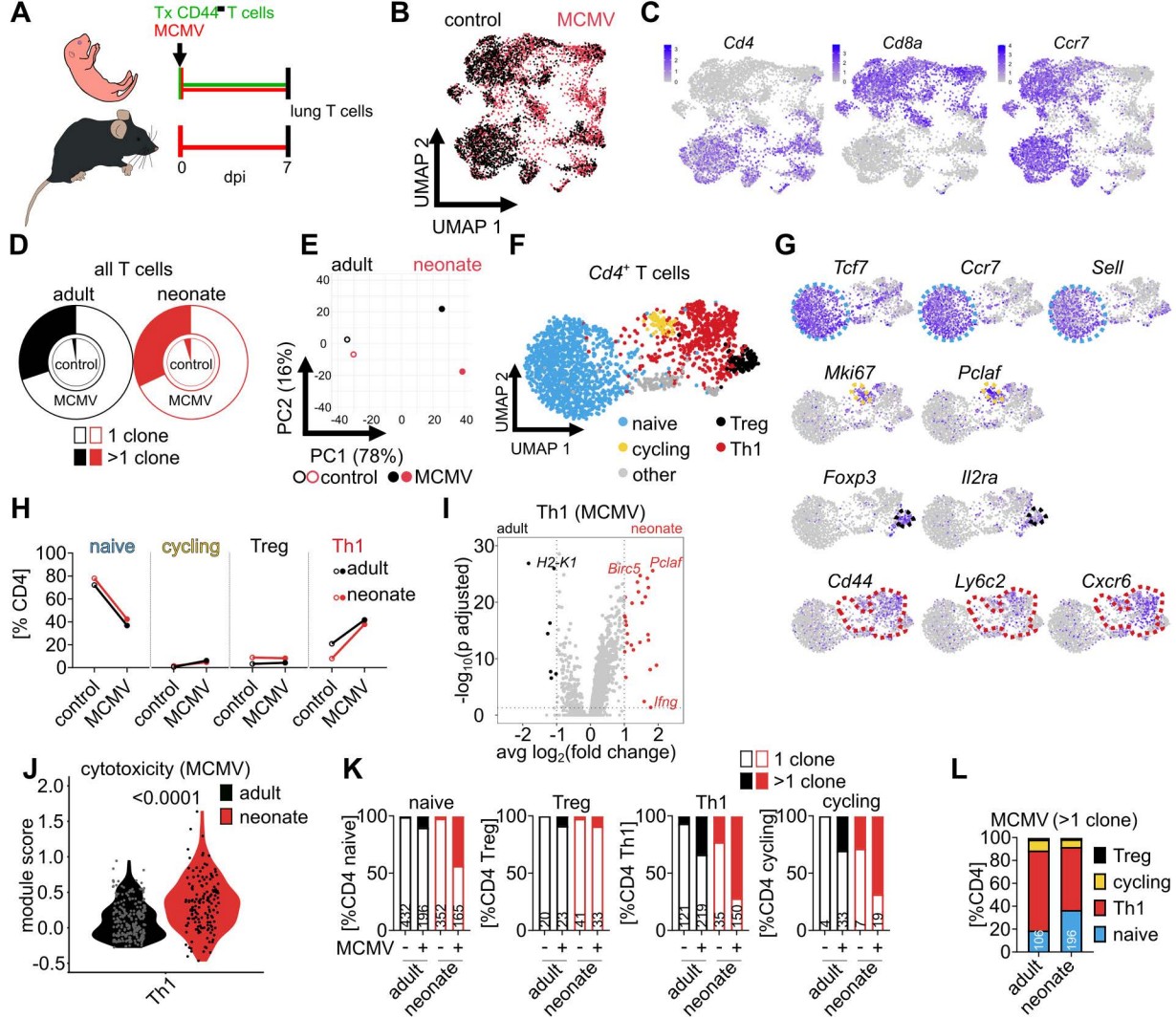

**Fig 3. CD4 T cells primed in neonates acquire a Th1-like phenotype with cytotoxic potential. (A)** Experimental setup for Figs 3 and 4: Neonatal (n = 3) and adult (n = 3) mice were infected with $10^4$ or $2 \times 10^5$ PFU MCMV, respectively. $10^7$ purified CD3+CD44- naïve T cells obtained from adult eGFP mice were transferred into neonatal mice. Control animals (n = 3 neonates and n = 3 adults) were not infected with MCMV. After 7 days, adoptively transferred eGFP+ lung T cells in neonates and endogenous lung T cells from adults were isolated, and pooled for single-cell transcriptome profiling (scRNAseq) combined with TCR sequencing. **(B)** Unsupervised RNAseq analysis and UMAP dimensionality reduction of all cells, colours indicate infection status. **(C)** Cell expression of genes as indicated. **(D)** Relative distribution of T cells with identical TCRs indicating clonal enrichment, the filled area represents clonal cells. **(E)** Principal component analysis for the four different groups. **(F)** Subanalysis and UMAP dimensionality reduction of *Cd4+Cd8a-* T cells. **(G)** Classification of CD4 T cells into subpopulations by expression of characteristic genes as illustrated. **(H)** Frequency of the four groups assigned to the different CD4 T cell subpopulations. **(I and J) (I)** Differential gene expression and **(J)** cytotoxicity module score of Th1 cells obtained from MCMV-infected adults and neonates. **(K)** Relative distribution of the clonal (filled section of bars) and non-clonal (open bars) CD4 T cell subpopulations of each group, with numbers indicating the absolute cell counts. **(L)** Relative distribution of clonally expanded CD4 T cells into subpopulations. Data acquired by one experiment with n = 3 animals per group. Statistical difference in **(J)** was calculated with Mann-Whitney U test, and the p-value is provided above the graph.

MCMV-mediated inflammatory conditions at 5 dpi. In general, more cytokines could be detected in non-infected adults than in neonates (Fig 5A - 5L). MCMV infection induced a shift in the local adult cytokine environment to higher concentrations of Interleukin-2 (IL-2), IL-7, and IL-15, which was not detected in neonates (Fig 5A - 5D). Moreover, there was an

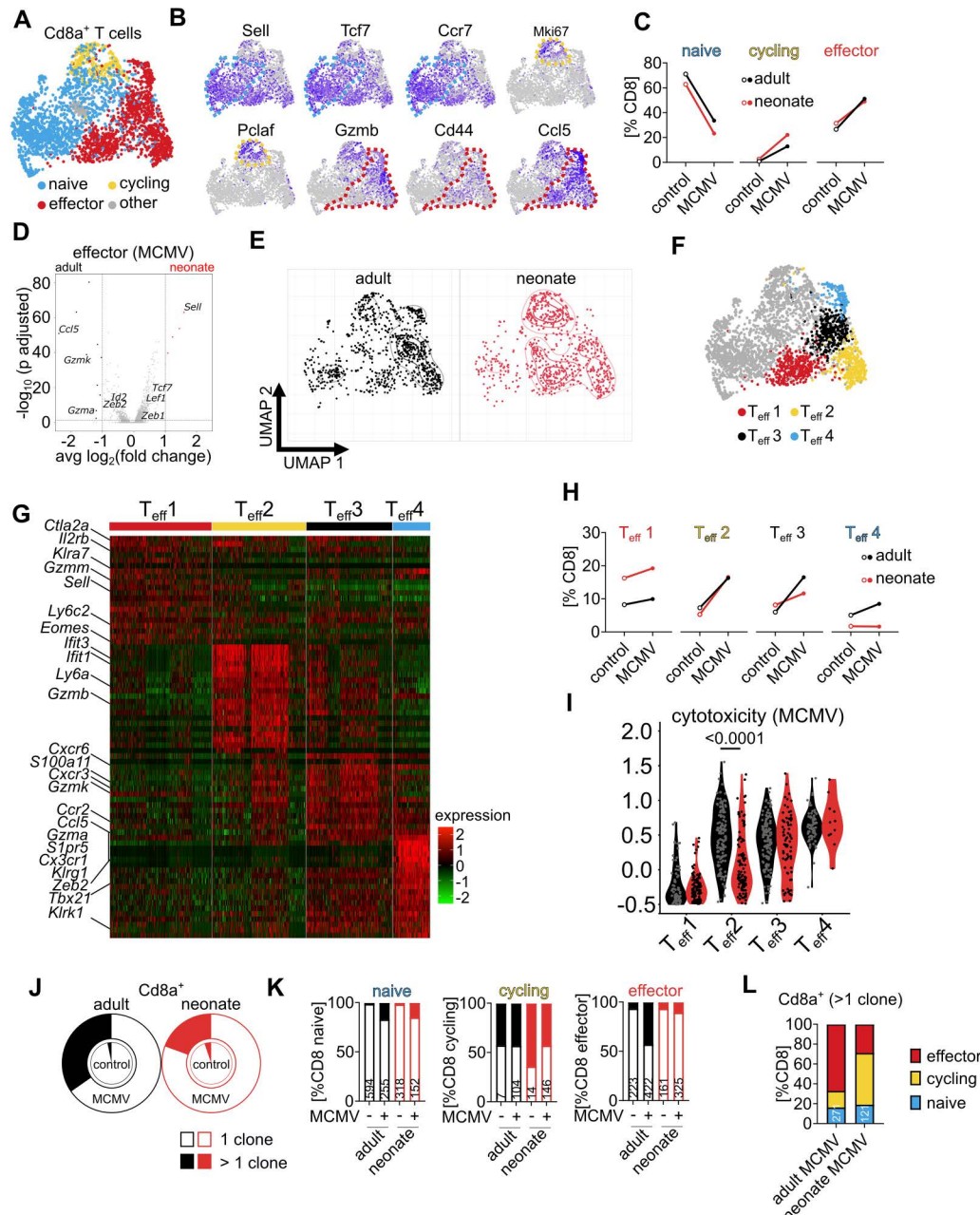

**Fig 4. CD8 T cells primed in neonates do not acquire an antiviral CTL phenotype. (A)** Subanalysis and UMAP dimensionality reduction of *Cd8a*+*Cd4*- T cells. **(B)** Classification of CD8 T cells into subpopulations by expression of characteristic genes as illustrated. **(C)** Frequency of the four groups assigned to the different CD8 T cell subpopulations. **(D)** Differential gene expression of effector CD8 T cells isolated from infected neonates and adults. **(E)** UMAP dimensionality reduction of CD8 T cells isolated from MCMV-infected animals. **(F)** Subclassification of effector CD8 T cells clusters. **(G)** Heatmap of top 20 highly expressed signature genes in CD8 T cell effector subclusters. **(H)** Frequency of the four groups assigned to the different CD8 effector T cell subpopulations. **(I)** Cytotoxicity module score of CD8 effector T cell subpopulations from MCMV-infected animals. **(J)** Relative distribution of CD8 T cells with identical TCRs indicating clonal enrichment, the filled area represents clonal cells. **(K)** Relative distribution of the clonal (filled section of bars) and non-clonal (open bars) CD8 T cell subpopulations of each group, with numbers indicating the absolute cell counts. **(L)** Relative distribution of clonally expanded CD8 T cells into subpopulations. Data acquired by one experiment with n = 3 animals per group. Statistical difference in **(I)** was calculated with Mann-Whitney U test, and the p-value is provided above the graph.

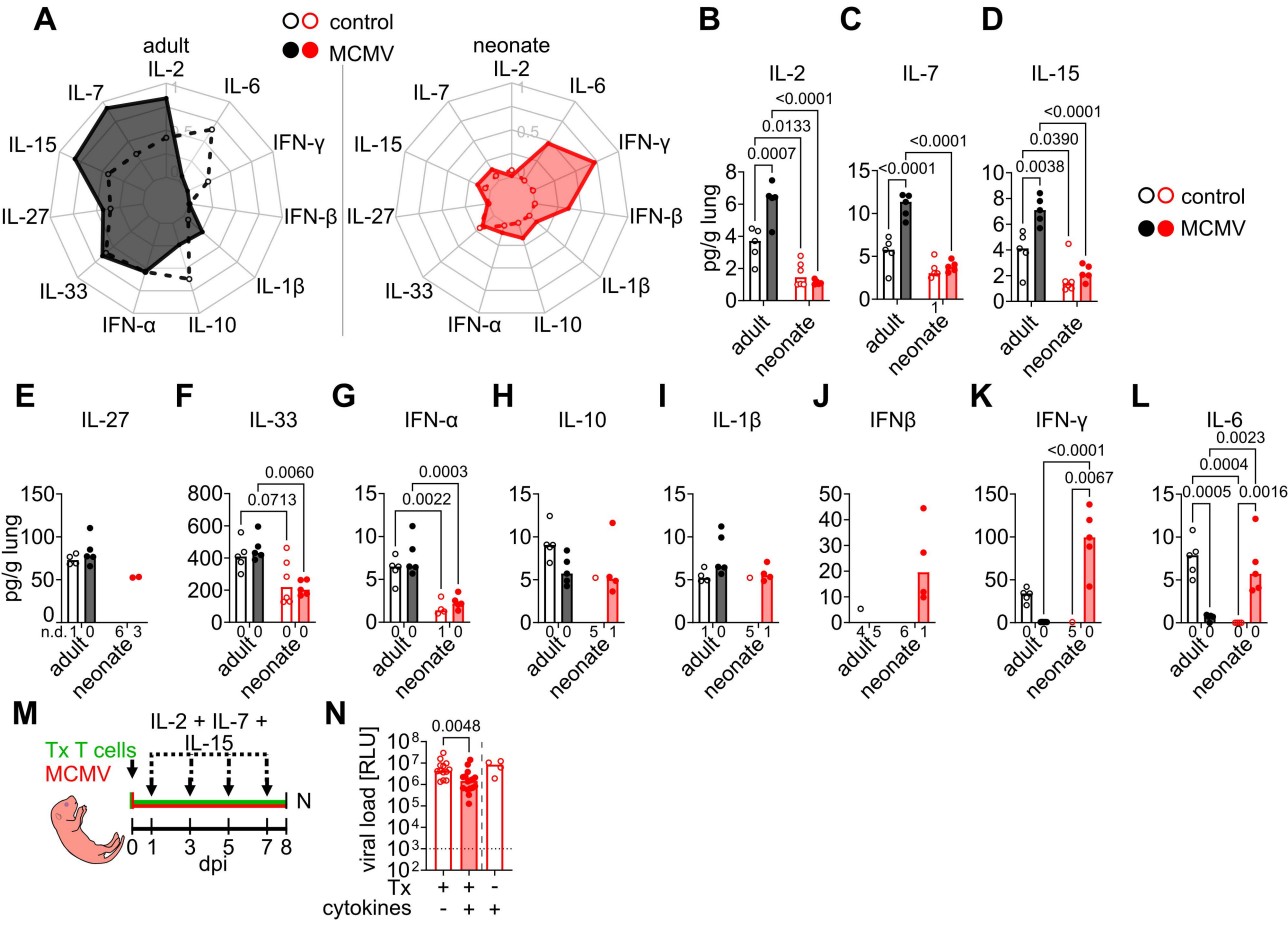

**Fig 5. Alterations in priming conditions in the early-life phase. (A-L)** Relative concentration of cytokines in steady state and MCMV-infected lungs at 5 dpi as indicated. **(M)** Experimental setup for **(N)**. MCMV-infected neonatal mice received $10^7$ eGFP$^+$ T cells and IL-2, IL-7, and IL-15 cytokine complexes or buffer control as indicated. **(N)** Lung viral loads after adoptive T cell transfer and treatment with cytokines or buffer control. Data acquired from two **(A-L)** or six **(M-N)** independent experiments. Values in A are shown on a normalized scale. Statistical differences were calculated with 2-way ANOVAs **(B-L)** or Mann-Whitney U test **(N)** and the p values are provided above the graphs. Numbers below columns of **(C-K)** indicated samples were the indicated cytokines was not detected (n.d.).

age-related difference for IL-27, IL-33, and Interferon-α (IFN-α), which were found in higher concentrations in adults (Fig 5E - 5G). In neonates, IL-10, IL-1β, and IFN-β were detected in MCMV-infected but not control lungs, and there was no significant difference compared to cytokine concentrations in adults (Fig 5H - 5J). IFN-γ as well as IL-6 were significantly higher in inflamed neonatal lungs than in controls (Fig 5K and 5L). Adult T cells adoptively transferred into MCMV-infected neonates together with treatment of cytokine-antibody or -receptor complexes of IL-2 [29], IL-7, and IL-15 [30], but not cytokine complexes alone, reduced lung viral loads (Fig 5M and 5N). Thus, there were remarkable age-related differences in signal 3 cytokines that likely impact T cell priming and effector differentiation.

## MCMV-primed adult effector T cells protect from early-life MCMV lung infection

To confirm that T cells primed in adults are more protective against MCMV we isolated CD44$^+$ T cells from infected adult lungs and adoptively transferred these into neonatal mice (Fig 6A). At 8 dpi we found a 1.9-fold higher frequency of T cells in lungs of mice that received adult effector T cells as compared to mice without cell transfer

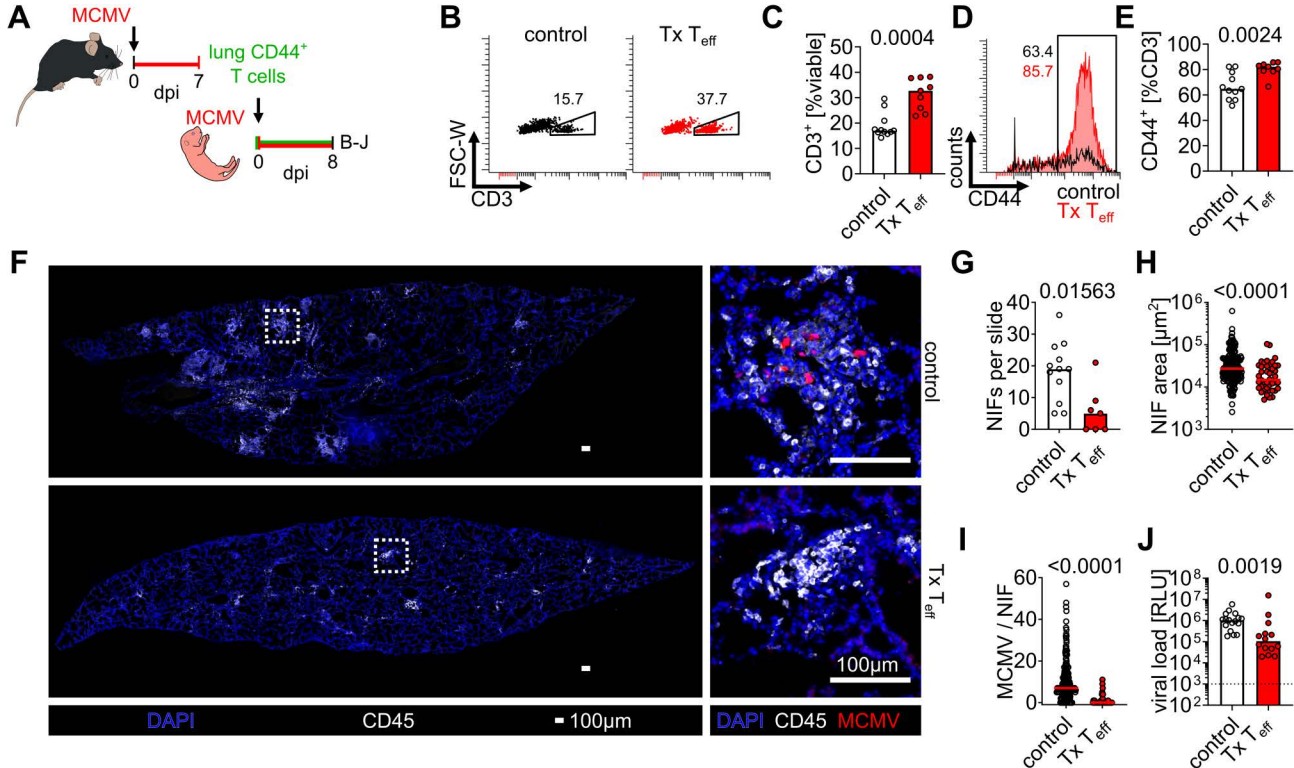

**Fig 6. Effector T cells primed in adults protect from early-life MCMV lung infection. (A)** Experimental setup for **(B-J)**: CD3⁺CD44⁺ effector T cells were purified from adult lungs 7 dpi, and 8x10⁶ cells per animal were transferred into MCMV-infected neonatal mice, which were then analysed 8 dpi. Controls did not receive T cells. **(B and C) (B)** Representative flow cytometry plots and **(C)** quantitative analysis of CD45⁺ cells obtained from neonatal lungs at 8 dpi. **(D and E) (D)** Representative flow cytometry histograms and **(E)** quantitative analysis of CD44 expression in isolated lung T cells. **(F)** Immunohistology of lungs in general view (left panels) and high resolution (right panels) illustrating inflammation and presence of NIFs with MCMV-infected cells. Dotted squares indicate the zoomed areas shown in the right panels. **(G-H)** Quantitative analysis of immunohistology with **(G)** number of NIFs per lung slice, **(H)** NIF area, and **(I)** number of MCMV-infected cells per NIF. **(J)** Quantitative analysis of lung viral loads as indicated. Data acquired from four independent experiments. Statistical differences were calculated in **C**, **E**, and **G-J** with Mann-Whitney U test, and the p-values are provided above the graphs.

(Fig 6B and 6C). The frequency of CD44⁺ cells was higher on T cells in mice that received the adoptive transfers, indicating that these cells had maintained their effector phenotype (Fig 6D and 6E). Adoptive transfer of effector T cells reduced inflammation in lungs (Fig 6F), defined by a reduced number and size of NIFs (Fig 6F - 6H), and the number of MCMV-infected cells within NIFs (Fig 6F and 6I). In line, the overall virus load was lower in the lungs of mice that received the T cell treatment (Fig 6J). Accordingly, the higher cytotoxicity score found in adult-primed CD8 T effector cells correlated with protective function, and these cells could reduce MCMV infection in neonatal mice.

## Discussion

Upon viral infection, CD8 T cell differentiation is shaped by multiple stimuli during the priming phase including TCR peptide binding, co-stimulation, and cytokine signaling. Integration of these stimuli drives the expression of several transcription factors, chemokine receptors, and effector molecules that affect the generation and the effector function of antiviral CTLs. In this study, we found distinct priming conditions to lead to significant differences in the phenotype and function of virus-specific CTLs in early life.

Murine respiratory tract CD8 effector T cells encompassed four subpopulations that differed in their expression of additional effector molecules. Among these, $T_{eff}3$ and $T_{eff}4$ were high in expression of *Gzmk*, *Gzma*, *Cx3cr1*, *S1pr5*, *Klrg1*, *Cxcr6*, and *Zeb2,* arguing for their late differentiation state [31]. Together with $T_{eff}2$, these three subpopulations exhibited a high cytotoxicity score upon adult MCMV challenge and are likely predominantly involved in the control of lung infection. In contrast, $T_{eff}1$ exhibited signatures of less differentiated effector T cells and had a low cytotoxicity score. With these criteria, approximately 40% of CD8 T cells primed in adults could be classified as CTL with a high cytotoxicity score and potential for anti-MCMV effector function *in situ*. This was contrasted by the fact that CD8 T cells primed in early life were less likely to acquire a $T_{eff}3$ or $T_{eff}4$ phenotype, and those cells that clustered as $T_{eff}2$ exhibited a low cytotoxicity score, resulting in an overall CTL-like phenotype of only ~10% of CD8 T cells. Thus, the superior capacity to control MCMV infection in adults is likely due to the observed higher frequencies of cytotoxic CTLs upon infection. Accordingly, effector CD8 T cells isolated from adults efficiently reduced MCMV loads in neonates. These data indicate that potential inhibitory mechanisms in neonates that would interfere with the function of adoptively transferred antiviral effector CD8 T cells are not prominent during the T cell effector phase in the young host. Instead, CD8 T cell priming in early life impairs their differentiation program into cytotoxic CTLs and increases their vulnerability to infection.

We found significantly higher production of IL-2, IL-7, and IL-15 in MCMV-infected adult lungs, whereas these cytokines were low in neonatal mice and did not increase in concentration in response to infection. Treatment with cytokine complexes resulted in reduced viral loads in neonates that received adoptive transfer of adult naïve T cells. Importantly, these three cytokines belong to the common receptor γ-chain (γc) family and are well-known for their roles in T cell proliferation, homeostasis, and differentiation [32]. IL-2 is known to drive CD8 effector T cell function and expression of cytotoxic molecules [33,34]. Moreover, protection by adoptive transfer of CD8 effector T cells into MCMV-infected irradiated adult mice was increased when combined with the application of recombinant IL-2 [35]. IL-7 is a critical survival factor for T cells and important for generating CD8 T cell memory [36]. *IL-15ra-/-* mice are deficient in CD8 T cells [37], and IL-15 is critical for the maintenance of effector memory CD8 T cells in infected lungs after MCMV infection [38]. Testing of cytokines involved in CD8 T cell polarization using the Immune Dictionary [39] revealed that IL-15, IL-2, and IL-7 ranked first, second, and fifteenth, respectively, in their ability to induce expression of cytotoxicity module score genes. Furthermore, stimulation of CD8 T cells with IL-15 *ex vivo* upregulates gene expression of *Ccl5* and *Cxcr6* [40], and here, we found CD8 T cells to exhibit higher expression of these genes and proteins in adults. A previous study suggested that CCL5 itself may recruit CD8 T cells in MCMV-infected lungs [41] and CXCR6 was shown to position CTLs next to IL-15 trans-presenting dendritic cells [42]. Thus, the highly abundant γc cytokines present in adult lungs may fuel local antiviral effects, further explaining the observed age differences in MCMV control. In addition, IL-33, IFN-α, and IL-27 were more abundant in adult than in neonatal mice. IL-33 promotes antiviral CD8 T cell immunity [43], and IFN-α itself is known as a signal 3 cytokine affecting T cell activation, proliferation, and survival [44]. IL-27 has been shown to enhance CTL cytotoxic function [45], and IL-27 receptor deficiency led to impaired expansion of potent anti-EBV effector cytotoxic CD8 T cells in humans [46]. Chronically MCMV-infected *IL-27ra-/-* mice exhibit no prominent phenotype in the antiviral CD8 T cell pool, but the role of IL-27 in the early T cell priming phase has not been studied [47,48]. Together, several cytokines that were present in higher concentrations in adults are involved in driving protective antiviral CTL immunity, suggesting that a deficiency of these cytokines in the early-life T cell priming phase caused ineffective CD8 T cell responses.

T cells were activated in neonatal lungs, indicating that the presentation of MCMV antigens is functional in early life. Indeed, we previously found that NIFs contain various types of antigen-presenting cells (APC) and serve as priming sites for MCMV-specific CD8 T cells [13]. Recently, a subset of CD103+ APCs has been proposed to dampen CD8 T cell responses, leading to impaired pulmonary immunity in the first two weeks of life in neonates [49]. Thus, besides the lack of several signal 3 cytokines, direct APC–T cell interaction may further interfere with early-life generation of antiviral CTLs. Future studies should characterize both the cellular composition and functional properties of neonatal myeloid cells in response to MCMV infection.

In contrast to the CD8 T cell response, CD4 T cells did not reveal an age-related defect in effector differentiation. Rather, CD4 T cells primed in early life acquired a Th1-like phenotype with an even higher cytotoxicity score and increased expression of *Ifng*. Accordingly, we found more IFN-γ in MCMV-infected lungs in neonates than in adults, suggesting that this cytokine counterbalances the absence of CTLs. This matches a study where T cell-produced IFN-γ was required to contain MCMV lung infection in adults [16]. It is well-described that CD4 T cells are involved in CMV control by promoting antibody production and direct antiviral effects in tissue [50,51]. However, their contribution appears minor to that of CD8 T cells [52], and in this model, they could not compensate for the observed defect in CD8 T cell effector response. However, it is intriguing that the early-life T cell priming machinery selectively interferes with the generation of CTLs but not CD4 Th1 immunity, and this needs further investigation.

T cells are important for the control of CMV, and effector CD8 T cells are present in newborns after HCMV infection [10]. Neonatal mice lack αβ T cells at birth, and similarly, there are no αβ T cells in humans before the end of the 1st trimester of gestation [53]. The number of T cells gradually increases during gestation, supporting the hypothesis that the inverse correlation of gestational age at HCMV infection and risk of symptomatic congenital CMV [54–56] is linked to the number of T cells present in the unborn fetus. In addition to that, the CDR3 regions of TCRs exhibit lower variations due to the low expression of TdT in early life [57,58]. Thus, a low number of T cells that can respond to CMV antigens, together with altered priming conditions, may lead to an ineffective antiviral T cell response and therefore contribute to the high susceptibility to CMV early in life. This study indicates that in-depth phenotypic and functional profiling of CMV-responsive CD8 T cells may be essential for accurately predicting the protective efficacy of CMV immunity in early-life infection.

In summary, we investigated the kinetics of MCMV-specific T cell responses in early life by delineating the expansion of endogenous CD8$^+$T cells and assessing the impact of adoptive transfer of naïve adult T cells into neonates. We characterized the mechanisms underlying the generation of non-cytotoxic CD8$^+$T cells, as well as whether impaired antiviral responses arise during priming or at the effector stage. Our data indicate that deficiencies in inflammatory cytokines during neonatal priming drive the emergence of non-cytotoxic CD8$^+$T cells, whereas adult-primed T cells retain full effector functionality in neonatal hosts, identifying the priming environment as the key determinant of the impaired response.

## Materials and methods

### Ethics statement

Animal experiments were performed according to the guidelines of the Federation of European Laboratory Animal Science Associations (FELASA) and Society of Laboratory Animals (GV-SOLAS) and approved by the institutional review board and by local authorities (Behörde für Gesundheit und Verbraucherschutz, Amt für Verbraucherschutz, Freie und Hansestadt Hamburg, reference numbers 06/16, 45/19, 04/20, and 45/24).

### Animals

C57BL/6J mice were purchased from Charles River Laboratories and bred in individually ventilated cages under specific pathogen-free conditions at the animal facility of the University Medical Center Hamburg-Eppendorf. β-actin-eGFP [59], β-actin-eCFP [60], *Rag2$^{-/-}$IL2rg$^{-/-}$* [61,62] and TCR transgenic OT-I [63] and OT-II [64] mice were all kept from a C57BL/6 background. OT-I mice were crossed with β-actin ECFP to generate OT-I$_{CFP}$ mice and OT-II mice with β-actin EGFP to generate OT-II$_{GFP}$ mice. Both male and female mice were used for this study. Neonatal mice within 24h of birth and adult mice from 6 to 36 weeks of age were used.

### Viruses

Infection experiments were performed with MCMV-3DR [14] if not assigned differently. This recombinant consists of an *mCherry*-P2A-*Gaussia luciferase* insertion within the m157 ORF [17] and additionally carries a sequence within the m164 ORF encoding the SIINFEKL peptide [65]. A previously described mutation with the m129 ORF has been repaired

allowing full-length expression of MCMV-encoded chemokine 2 (MCK2) [19]. MCMV-4DR is a modification of MCMV-3DR, where the sequence for a fusion protein made up of the first 118 residues of the human transferrin receptor linked to residues 299–385 from chicken ovalbumin was inserted into the *m128* ORF under control of the human major immediate early promotor (Fig 2J). MCMV-2DR [18] does not encode for ovalbumin peptides. All recombinants were derived from the pSM3fr Smith strain and modified by bacterial artificial chromosome mutagenesis using the *en passant* protocol [66], propagated and titrated on 10.1 [67] and M2-10B4 (ATCC CRL-1972) fibroblasts, respectively.

## Infections

Neonatal mice were infected intratracheally within the first day of life (post-natal day 0) with $10^4$ PFU MCMV diluted in 10 µL PBS. Adult (6–36 weeks) mice were infected with $2x10^5$ PFU ($10^6$ for experiments depicted in Fig 1) intranasally after anesthesia (100 mg/kg body weight ketamine and 5mg/kg body weight xylazine, intraperitoneally).

## Adoptive T cell transfers

Polyclonal T cells were isolated from spleens and lymph nodes by negative selection using a Pan T Cell Isolation Kit II (Miltenyi Biotec) according to the manufacturer's guidelines. Naïve $CD44^-$ T cells were isolated by magnetically depleting CD44+ cells using CD44 Microbeads (Miltenyi Biotec). $CD44^+$ T cells from non-infected mice were isolated by collecting the positive fraction using CD44 Microbeads (Miltenyi Biotec). Untouched OT-I and OT-II cells were isolated using a $CD8a^+$ and $CD4^+$ T cell Isolation Kit (Miltenyi Biotec), respectively. Stainings with Cell Proliferation Dye eFluor 450 or 670 (eBioscience) were performed according to the manufacturer's instructions.

CD44+ T cells were isolated from lungs of MCMV-infected *Wt* adult mice 7 dpi by positive selection of $CD45^+$ cells with CD45 Microbeads (Miltenyi Biotec) followed by FACS cell sorting for $CD3^+CD44^+$ T cells. Cells were injected intraperitoneally at PND 0.

## Cytokine treatment

Cytokine complexes were freshly prepared by pre-incubation of IL-2 and αIL-2 (clone S4B6), IL-7 and αIL-7 (clone M25), and IL-15 and IL-15Rα at a 1:2 molar ratio for 20 min at 37°C as previously described [29,30]. Cytokine complexes were then pooled, and 75ng of each cytokine was injected i.p. into neonates.

## Leukocyte isolation from lungs

Leukocytes were isolated after lung digestion using a Lung Dissociation Kit (Miltenyi Biotec) according to the manufacturer's guidelines. Cells were then isolated via magnetic isolation or cell sorting as specified in the experiments.

## Histology

Organs were fixed in 2% PFA containing 30% sucrose, embedded in Tissue-Tek O.C.T. medium and kept at -20°C until further processing. Cryosections of 7µm (slide scanner) or 10µm thickness (confocal microscopy) were stained after Fc receptor blocking with CD3-Alex Fluor 647 (17A2), B220-PE (RA3-6B2), CD45-APC or CD45-Alexa Fluor 750 (30-F11) and DAPI. Images were acquired with a ZEISS Axioscan 7 Microscope Slide Scanner or with a Leica TCS SP8 confocal microscope. Images were processed with ZEN 2.6 or LAS AF Lite 4.0, respectively.

## Quantification of histology data

Infected cells and leukocytes from lungs were identified based on fluorophore expression (MCMV mCherry, OT-$I_{CFP}$, and OT-$II_{GFP}$), antibody or DAPI staining, and manually counted. In the adoptive transfers of OT-$I_{CFP}$ and OT-$II_{GFP}$ T cells, three to four NIFs per animal were analysed. In the remaining experiments, one whole lung section per animal was analysed.

NIFs were identified based on preferential accumulation of CD45[+] cells and/or nuclei, manually defined, and areas calculated with ZEN 2.6.

## Flow cytometry, intracellular staining and cell sorting of mouse samples

Lung single cell suspensions were prepared as described above. Spleens and lymph nodes were mashed through 70μm cell strainers to obtain single cell suspensions and blood leukocytes were obtained after erythrocyte lysis for 15 min at room temperature. Cell surface stainings were then performed after blocking of Fc receptors with CD3-FITC, CD3-Brilliant Violet 711, CD3-Alexa Fluor 647 (17A2), CD3-PE (REA641)), CD4-PE-Vio770, CD4-PerCP (REA604), CD8b-APC (YTS156.7.7), CD8b-APC-Vio770 (REA793), CD44-APC, CD44-PE, CD44-Pacific Blue (IM7), CD44-PE-Vio770 (REA664), CD45-APC (30-F11), CD45R/B220-FITC (REA755), CD62L-Alexa Fluor 488 (MEL-14), CXCR6-Brilliant Violet 711 (SA051D1), KLRG1-APC-Cy7(2F1/KLRG1), NK1.1-APC (PK136), TCRvα2-PE (B20.1), TCRβ-PerCP-Vio700 (REA318) antibodies, M38-, M45-, m139-PE tetramers (provided by Ramon Arens, Leiden University, The Netherlands) or M25-PE tetramer (NIH tetramer facility). For intracellular stainings, cells were fixed for 30 min after surface staining and incubated with CCL5-PE (2E9/CCL5) in permeabilization buffer. Single cell suspensions were measured within one day on a BD FACSCanto II or BD FACS LSRFortessa flow cytometer. Isolation by cell sorting was performed on a BD FACSAria Fusion.

## Cytokine multiplex assay

Mouse lungs were collected at 5 dpi after perfusion with PBS, weighed, and lysed with a TissueLyser II (Qiagen) in 200μL PBS. Supernatants were frozen at -80°C until further processing. Supernatants were thawed on ice and processed with a LEGENDplex (Biolegend) custom made panel targeting IFN-α, IFN-β, IFN-γ, IL-10, IL-12p70, IL-15, IL-18, IL-1β, IL-2, IL-27, IL-33, IL-6, IL-7 and TGF-β1 (free active) according to the manufacturer's guidelines. Samples were then measured on a BD FACSCanto II. Data analysis was performed on the LEGENDplex Data Analysis Software Suite (Biolegend).

## Determination of viral loads by MCMV-encoded luciferase activity

Organs were homogenized with a TissueLyser II (Qiagen) machine in phosphate buffered saline, and MCMV-3DR-encoded *Gaussia* luciferase activity in organ supernatants after centrifugation was measured in duplicates after automatic addition of Coelenterazine (Synchem) with a Centro LB 960 XS3 microplate luminometer (Berthold). Luciferase values from non-infected animals were used as controls and/or to determine organ-specific limit of detection.

## Single cell RNA sequencing

Adult *Wt* mice were infected with $2x10^5$ PFU MCMV-3DR. Naïve CD44[-] T cells were isolated from age-matched eGFP mice using a combination of a magnetic cell sorting Pan T cell kit (Miltenyi Biotec) and subsequent CD44[+] depletion as described above. After isolation, $10^7$ eGFP naïve CD44[-] T cells were adoptively transferred into PND 0 neonates, which were simultaneously infected with $10^4$ PFU MCMV-3DR. At 7dpi, adult and neonatal mouse lungs were collected after perfusion with PBS, and single cell suspensions were generated as described above. Lung T cells were then isolated by positive selection of T cells using CD3ε Microbeads (Miltenyi Biotec). Isolated cells were then stained with a CD3 antibody and FACS-sorted for CD3[+] cells (adults) or CD3[+]eGFP[+] cells (neonates). Single-cell RNA-seq libraries were constructed using a Chromium Single Cell 5' Kit (v2) according to the manufacturer's instructions.

## Single cell RNA-seq data analysis

### Data processing

Raw FASTQ files were generated from Illumina sequencer's base call files (BCLs) using the 10x Genomics Cell Ranger (version 4.0.0) [68] mkfastq pipeline. Count matrices of valid barcodes and feature/gene expression were generated by

the Cell ranger count pipeline using mouse reference genome GRCm38 and GENCODE gene annotation version M23 provided by 10x Genomics. Custom GFP genome and annotation (using "MT-" as a prefix) were added to the mouse genome, as well as to the mouse annotations, respectively.

## Creating single-cell objects and filtering

Filtered count matrices generated above for each sample were taken as input in Read10x function in R package Seurat (v4.2.0), and used for further downstream analysis. Following filters were used to create a scRNA-seq object using CreateSeuratObject function for each sample: 1) genes were removed if detected in less than 10 cells; 2) cells with a total number of genes below 200 were removed; 3) cells with a total number of genes detected above the 99th percentile were removed; 4) cells with a proportion of mitochondrial UMIs of more than 5% were removed; and 5) cells with total UMIs less than 1000 were removed.

Separate datasets were created of the object created above for *Cd8a* positive cells and *Cd4* positive cells, and the following strategies were performed in a similar way for sub-clustering as the original integrated scRNA-seq object with all samples.

## Normalization, integration and downstream analysis

Normalization and variance stabilization of each scRNA-seq object was performed using the statistical modelling framework sctransform [69] implemented in Seurat, and also removing confounding sources of variation such as: 1) total UMIs in a cell, 2) mitochondrial mapping percentage, and 3) eGFP expression. Resulting objects for all samples were integrated step by step using Seurat functions in order: SelectIntegrationFeatures (with parameter nfeatures = 3000); PrepSCTIntegration; FindIntegrationAnchors, and IntegrateData. Dimension reduction of the integrated object created was performed using the RunPCA function, and used as input to perform non-linear dimension reduction by using UMAP (Uniform Manifold Approximation and Projection) method with the function RunUMAP using 30 dimensions. Prior to identifying clusters of cells, a shared nearest neighbor (SNN) graph was constructed using FindNeighbors function, and used as input for clustering by Louvain algorithm implemented in the FindClusters function.

Default assay of the integrated object was switched to RNA and normalized using NormalizeData function to plot gene expression values. Default assay was switched back to integrated, and FindConservedMarkers function using default parameters was used to identify marker genes for all clusters of an integrated object. Cluster type identification and re-naming were done manually based on the expression of the genes described in the results section.

Heatmap of top 20 marker genes in order of log2FC was created using the DoHeatmap function. Cytotoxicity scores were added with the AddModuleScore including *Fasl*, *Gzma*, *Gzmb*, *Gzmc*, *Gzmk*, *Gzmm*, *Tnf*, *Tnfsf10*, *Prf1* and *Ifng* as features [70,71].

## TCR repertoire analysis

FASTQ files generated above were used with Cell Ranger vdj pipeline to perform V(D)J transcripts assembly and calling paired clonotypes, using V(D)J mouse reference (vdj-GRCm38-alts-ensembl-4.0.0) provided by 10x Genomics. Filtered contig annotation files for each sample from the pipeline were used to combine with clonotypes file using R script [72] to create TCR repertoire metric files for each sample containing information such as: cell barcode; TCR clonotype IDs; v_gene; j_gene and CDR3 sequence. Clonotype IDs output by the Cell Ranger vdj pipeline were used to define clonotype frequency if more than one cell was represented by it.

scRNA-seq and TCR repertoire analysis, plots and tables were generated in R version 4.1.1 with ggplot2 version 3.3.6 [73].

## Pseudo-bulk RNA-seq analysis

A gene expression matrix was created using AggregateExpression function on the counts slot of each scRNA-seq object. DESeq2 [74] version 1.32.0 was used for normalization and to perform pairwise differential expression analysis.

## Statistical analyses

Statistical analyses were performed with GraphPad Prism 8 using the statistical tests as indicated. Radar charts were created using the function radarchart from the package fmsb [75].

## Supporting information

**S1 Data. Raw data file for Figs 1–6 and S1-4.**
(XLSX)

**S1 Fig. Detection of MCMV-specific CD4 T cells and accumulation of T and NK cells in spleens of non-infected mice.** Related to Fig 1. (A and B) (A) Representative flow cytometry plots and (B) pooled analysis of M25-specific CD4 T cells in the blood of infected and non-infected control mice. (C) Accumulation of NK cells, CD4 and CD8 T cells in the spleen (data as depicted in Fig 1F, left panel but with adjusted y-axis scale). Data display pooled results from 2 or more independent experiments (A and B, n = 2–14 per time point, C, n = 3–7 per time point). Lines in (B) indicate the median value. Numbers above each graph in (B) indicate the p value between adults and neonates of a 2-way ANOVA.
(PDF)

**S2 Fig. Adoptive transfer of adult naïve T cells into neonates is not protective against MCMV.** Related to Fig 2. (A) Quantitative analysis of CD44 expression on adoptively transferred lung eGFP⁺ T cells. Related to Fig 2D - 2G. (B) Quantitative analysis of lung immunohistology after adoptive transfer of adult polyclonal T cells. Related to Fig 2H. (C) Representative immunohistology of a neonatal lung NIF after MCMV-2DR infection and adoptive transfer of OT-I and OT-II cells. (D) Number of OT-I and OT-II cells in neonatal NIFs after MCMV-2DR or MCMV-4DR infection. (E) Correlation of number of OT-I cells per NIF with number of MCMV-4DR-infected cells per NIF in neonates. (E) NIF area and number of infected cells in NIFs of neonates infected with MCMV-4DR. Data display pooled results from 3 or more independent experiments (A, n = 5, B, n = 7–11 per time point, C-E, n = 3–4 per group). Numbers above each graph in (A), (C), and (E) indicate the p values of Mann-Whitney tests. (D) Nonparametric Spearman correlation r and two-tailed p value is provided.
(PDF)

**S3 Fig. Single-cell RNA sequencing of T cells primed in adult and neonatal mice.** Related to Fig 3. (A) Absolute numbers of analysed T cells in each group. (B) UMAP dimensionality reduction of CD4 T cells isolated from MCMV-infected animals. (C) Relative distribution of clonal sizes of all CD4 T cells. (D) Cytotoxicity module score of CD4 Th1 cells isolated from non-infected adults and neonates.
(PDF)

**S4 Fig. Effector phenotype of CD8 T cells primed in adult and neonatal mice.** Related to Fig 4. (A) Analysis of CD44 protein expression by flow cytometry in adoptively transferred (eGFP⁺) or endogenous CD8⁺ T cells of neonates and adults, respectively. (B and C) (B) Representative flow cytometry plot and (C) pooled analysis of CXCR6 and CCL5 expression in adoptively transferred (eGFP⁺) or endogenous CD8⁺CD44⁺ T cells of neonates and adults, respectively. (D) Cytotoxicity module score of effector CD8 T cells isolated from non-infected adults and neonates. Data in A-C display pooled results from 2 independent experiments (n = 5–6). Numbers above each graph in (A) and (C) indicate the p values of 2-way ANOVAs.
(PDF)

**S1 Table. Differentially expressed genes of Th1 CD4 cells isolated from MCMV-infected adult and neonatal mice.** Related to Fig 3.
(XLSX)

**S2 Table. Differentially expressed genes of effector CD8 cells isolated from MCMV-infected adult and neonatal mice.** Related to Fig 4.
(XLSX)

## Acknowledgments

We thank Martin Messerle for providing MCMV-2DR and Hans-Willi Mittrücker for critically reading the manuscript and discussion. We thank the NIH Tetramer facility for providing M25 MHC-II tetramers. Data acquisition was performed using instruments of the University Medical Center Hamburg-Eppendorf (UKE) Microscopy Imaging Facility and the UKE Cytometry and Cell Sorting Core Unit. The Leibniz Institute of Virology is supported by the Free and Hanseatic City of Hamburg and the Federal Ministry of Health.

## Author contributions

**Conceptualization:** Luís Fonseca Brito, Felix Rolf Stahl.

**Data curation:** Sanamjeet Virdi.

**Formal analysis:** Luís Fonseca Brito, Felix Rolf Stahl.

**Funding acquisition:** Wolfram Brune, Felix Rolf Stahl.

**Investigation:** Luís Fonseca Brito, Eleonore Ostermann, Silvia Tödter, Daniela Indenbirken, Renke Brixel, Adam Grundhoff, Felix Rolf Stahl.

**Methodology:** Felix Rolf Stahl.

**Resources:** Ramon Arens, Wolfram Brune.

**Supervision:** Wolfram Brune, Felix Rolf Stahl.

**Visualization:** Luís Fonseca Brito, Felix Rolf Stahl.

**Writing – original draft:** Luís Fonseca Brito, Felix Rolf Stahl.

**Writing – review & editing:** Luís Fonseca Brito, Wolfram Brune, Felix Rolf Stahl.

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
