## [Decision Letter · Decision Letter 0]

2 Feb 2026

Limited protection against early-life lung murine cytomegalovirus infection results from deficiency of cytotoxic CD8 T cells

PLOS Pathogens

Dear Dr. Stahl,

Thank you for submitting your manuscript to PLOS Pathogens. After careful consideration, we feel that it has merit but does not fully meet PLOS Pathogens's publication criteria as it currently stands. Therefore, we invite you to submit a revised version of the manuscript that addresses the points raised during the review process.

We look forward to receiving your revised manuscript.

Kind regards,

Eain A Murphy, Ph.D.

Academic Editor

PLOS Pathogens

Blossom Damania

Section Editor

PLOS Pathogens

Editor-in-Chief

PLOS Pathogens

PLOS Pathogens

orcid.org/0000-0002-7699-2064

**Additional Editor Comments:**

Dr. Stahl,

First of all let me begin by apologizing for the length of time it took to get reviews on your manuscript. Securing reviewers was a difficult process as invitees took days to inform us that they were too busy to provide a review and of the three reviewers we did secure, one failed to provide a review in a timely fashion in which their review was not received in over a month. I elected to drop them as a reviewer and make a decision with the two reviews we did receive. That being said, both reviewers were highly positive in terms of the quality and impact of your offering. Some concerns were raised (particularly with R1) which I believe can easily be addressed by your team. I anticipate the modified resubmission will be favorably recieved by the editorial staff.

sincerely, and again, I do apologize for the extended review time

Eain A. Murphy Ph.D.

**Journal Requirements:**

https://journals.plos.org/plospathogens/s/submission-guidelines#loc-parts-of-a-submission

- ® on page: 11

- TM on page: 12.

5) We have noticed that you have uploaded Supporting Information files, but you have not included a list of legends. Please add a full list of legends for your Supporting Information files after the references list.

Potential Copyright Issues:

i) Figures 2A, 2J, 3A, 5M, and 6A. Please confirm whether you drew the images / clip-art within the figure panels by hand. If you did not draw the images, please provide (a) a link to the source of the images or icons and their license / terms of use; or (b) written permission from the copyright holder to publish the images or icons under our CC BY 4.0 license. Alternatively, you may replace the images with open source alternatives. See these open source resources you may use to replace images / clip-art:

7) In the online submission form, you indicated that Any additional information required to reanalyse the data reported in this paper is available from the lead contact upon request.. All PLOS journals now require all data underlying the findings described in their manuscript to be freely available to other researchers, either

1. In a public repository

2. Within the manuscript itself

3. Uploaded as supplementary information.

8) Please amend your detailed Financial Disclosure statement. This is published with the article. It must therefore be completed in full sentences and contain the exact wording you wish to be published.

2) If any authors received a salary from any of your funders, please state which authors and which funders..

9)  Please ensure that the funders and grant numbers match between the Financial Disclosure field and the Funding Information tab in your submission form. Note that the funders must be provided in the same order in both places as well.

**Reviewers' Comments:**

Reviewer's Responses to Questions

**Part I - Summary**

Reviewer #1: The authors provide convincing evidence for the delayed T cell response in MCMV-infected neonatal mice, and its implication in infection severity. The study uses various relevant and complementary approaches and techniques. Overall, the study is well done and convincing, and provides novel information that will be informative to the field. Some changes and corrections are however needed to improve the quality of the manuscript as currently presented.

Reviewer #2: In this study, Brito et al examine the mechanisms that make mice susceptible to lung disease caused by infection with the murine cytomegalovirus (MCMV) in the early stages of life. Through a series of well-executed experiments, including adoptive T-cell transfer, the researchers convincingly demonstrate that defective cytokine-mediated priming of MCMV-specific CD8+ T cells in neonates is a key factor for the increased risk of viral lung infection in early life.

The work is presented almost flawlessly, and all conclusions appear to be justified by the provided data.

**Part II – Major Issues: Key Experiments Required for Acceptance**

Please use this section to detail the key new experiments or modifications of existing experiments that should be absolutely required to validate study conclusions.required to validate study conclusions.required to validate study conclusions.required to validate study conclusions.

Reviewer #1: 1) Fig1a (and following figures). The authors do not explain how the viral load was quantified. I assume by plaque assay here, and maybe via IVIS in Fig.2i. Please mention it in the figure legends and material and method.

2) Fig1b-c (and S1a-b). n=3 mice per time point is low. An additional experiment with at least 3 more animals per time-point is needed to give more strength to the results.

3) “In contrast, there was a trend for more IL-10, IL-1β, and IFN-β (Fig. 5h-j) in neonates, and IFN-γ as well as IL-6 (Fig. 5k, l)…”. There is no statistical difference shown in Fig.5 between adult and neonates, not even a trend, regarding IL-10 and IL1b production during infection. Quite the opposite in fact, as IL10 and IL1b are present in non-infected adults, but not in neonates. Also, while the increase of IL-6 and IFNg seems to be true, can the authors explain/discuss why IL-6 and IFNg are decreased (even undetected) following infection in adult mice? Those values are somewhat unexpected. Is there not a group/condition inversion here in the way this result was presented? Finally, some numbers appear below the x-axis on Fig5.e, g, h, i, j, and k, and need to be removed.

4) An important general comment is that by the time of infection, there are virtually no T cells in the neonates (Fig1e). One might argue that priming is not entirely the problem, and that the lack of T cells overall plays a major role. While the adoptive transfer experiments provide some clues to this question, T cells are transferred at the same time of the infection. At this time, the majority of these cells will not localized in secondary lymphoid organs, a likely site of their priming.

Did the authors ever infected pups at ~PND5-6, when endogenous T cells are present? Is there still a delay in tetramer+ T cells appearance? Would T cell have a decreased expression of CD44 at this time? Such a time-point is still very relevant to neonatal infection, as the 1st week of life in mice roughly match to the 3rd trimester of gestation in humans, especially with regards to immune system maturation.

Reviewer #2: My only concern about this study is that the authors could further clarify what significant advances they are presenting compared to the existing literature. While the results are convincing and well presented, the findings are unsurprising and offer limited novelty.

**Part III – Minor Issues: Editorial and Data Presentation Modifications**

Reviewer #1: 1) Introduction: “…reasonable to assume that age-related differences in anti-CMV T cell immunity are responsible for the increased vulnerability in early life.". Involved, would likely be a more appropriate word. Most immune cells display drastic variation in neonates compared to adult mice, in number and/or phenotype.

2) Introduction: “The T cell effector phase itself was not disrupted indicating…”. Not clear what the authors refer to as effector phase. Priming is impaired, and with it, so are the effector functions of the T cells. This thought needs to be fleshed out.

3) FigS1a. m25-tetramer staining is not extremely convincing. This would benefit significantly from using double positive tetramer staining (e.g. Tet-APC+ Tet-PE+) to improve signal to noise ratio. Similar point for Fig2f (4 cells in the gate). This is likely not trivial in neonatal mice, but can the authors concatenate all of the samples from a group (on flowjo) to increase the number of cells shown in the dotplot/gate?

4) Fig2d. “found the adult T cells to acquire CD44 membrane expression and proliferate”. While proliferation is undeniable, CD44 “acquisition” is (based on the provided plot). A large proportion of CD3+ cells are CD44+ (eF450+) in control animals. The authors need to either quantify and show total CD44+ cells (to prove their claim of increased expression of CD44) or rephrase to not be misleading.

5) “… neonates can compensate for the deficiency of this cell population in the early-life period, lead to antigen-specific proliferation, gain of effector marker expression, homing into sites of MCMV infection, but has no influence on virus load in the lung.”. CD44 upregulation is not shown and likely weak. Without cytokines, cytotoxic molecules, or other membrane markers (41BB, OX40, CD25, CD69…), this is an overstatement.

6) “However, there was a trend that adult T cells primed in MCMV infected neonates were more likely to differentiate into Th1 cells” Unclear why the authors make this claim, as the percentage of “Th1” is the same in both group during MCMV infection. In a more general way, it is unclear why the authors focus on CD4 here, as figure 1 and 2 clearly point toward CD8 T cells. Finally, these results suggest higher priming efficiency in neonates, directly contradicting their previous claim. Some discussion or clarification would be welcome here. The CD4 T cell transcriptomic analysis might be better suited after the CD8 transcriptomic analysis and kept in supplementary data.

7) Fig4d. “CD62L-encoding gene Sell and differentiation-repressing genes Tcf7, Lef1, and Zeb1 were more abundant in neonates (Fig. 4d, Table S2).”. Please add gene names on the plot. Only Sell is shown.

8) Fig4h. Teff1, which has the lower cytotoxic score, is the major effector cluster in neonates (before and after infection). Teff2, which proportion is similar in adult and neonates, show a decreased cytotoxic score in neonates. Teff3 and 4 (High cytotoxic score) are decreased in neonates. Data are convincing. The results description would benefit by rephrasing. Also, “…indicating that true CTLs are found within clusters”, this is only a transcriptomic data, not functional, please rephrase “true CTLs”.

9) Fig6b ”At 8 dpi we found 1.9-fold more T cells in lungs”. Figure shows %, not absolute number. Need to rephrase, and/or to show the absolute number of T cells. “CD44 expression was higher on T cells in mice that received the adoptive transfers indicating that these cells had maintained their effector phenotype (Fig. 6d, e).” This is not actually shown. This claim implies increased MFI, which is not shown (and seems unchanged based on the plot shown). There is however an increase in the percentage of T cell expressing CD44.

The effect of the T cell transfer on the NIFs and viral load is very convincing.

An additional informative control here would be to inject effector T cell from non-MCMV infected mice. In other words, is the effector T cell transfer efficient at reducing viral load due to bystander action, or is it by MCMV-specific T cells? Did the authors stain transferred CD44+ T cell with tetramers before transfer and at endpoint? The authors could also get at this issue by transfering pre-activated OT-I and OT-II, and infect with a WT virus (not expressing OVAp)

10) Discussion.

“Murine respiratory tract Cd44-, Gzmb-, and Ccl5-expressing CD8 effector T cells encompassed four

subpopulations that differed in their expression of effector molecules.” Needs rephrasing here. Those markers are not the one allowing sub-selection of total CD8 effector T cells. Indeed as per Fig4b and g, only a fraction of effector CD8 expresses GzmB and CD44. “…and those cells that clustered as Teff2 exhibited..” please rephrase.

“Accordingly, effector CD8 T cells isolated from adults efficiently reduced MCMV loads in neonates

arguing that the CD8 T cell effector function is not disrupted in the young host.” This is not clear. It seems the authors want to say that there are no inhibitory mechanisms or pathway in neonates that would inhibit pre-existing effector function from the adult CD8 T cells. “Instead, CD8 T cell priming in early life impairs their differentiation program into cytotoxic CTLs”. It is likely that the priming itself is impaired, leading to defective differentiation

Reviewer #2: The manuscript is well written, with only the occasional typo (e.g. Author Summary: 'effector T cells obtain from adult hosts'; Results: 'there were remarkable differences in signal 3 cytokines'). These errors should be corrected.

PLOS authors have the option to publish the peer review history of their article (what does this mean?). If published, this will include your full peer review and any attached files.). If published, this will include your full peer review and any attached files.). If published, this will include your full peer review and any attached files.). If published, this will include your full peer review and any attached files.

...

Reviewer #1: No

Reviewer #2: No

**Figure resubmission:**

**Reproducibility:**



---

## [Editor Report · Decision Letter 1]

7 Apr 2026

Dear Professor Stahl,

We are pleased to inform you that your manuscript 'Limited protection against early-life lung murine cytomegalovirus infection results from deficiency of cytotoxic CD8 T cells' has been provisionally accepted for publication in PLOS Pathogens.

Best regards,

Eain A Murphy, Ph.D.

Academic Editor

PLOS Pathogens

Blossom Damania

Section Editor

PLOS Pathogens

Sumita Bhaduri-McIntosh

Editor-in-Chief

PLOS Pathogens

orcid.org/0000-0003-2946-9497

Michael Malim

Editor-in-Chief

PLOS Pathogens

orcid.org/0000-0002-7699-2064

Dear Dr. Stahl,

Thank you for your resubmitted manuscript to PLoS Pathogens. Your revised work has been reviewed by the Editorial Board who have come to the conclusion that this work is significantly improved with the modifications you have made and that the offering is now at a level that is suitable for publication in the current form. With that we are designating this submission as ACCEPTED without further external review. Congratulations on a nice body of work.

Cheers,

Eain Murphy Ph.D.
---

## [Editor Report · Acceptance letter]

Dear Professor Stahl,

We are delighted to inform you that your manuscript, "Limited protection against early-life lung murine cytomegalovirus infection results from deficiency of cytotoxic CD8 T cells," has been formally accepted for publication in PLOS Pathogens.

Best regards,

Sumita Bhaduri-McIntosh

Editor-in-Chief

PLOS Pathogens

orcid.org/0000-0003-2946-9497

Michael Malim

Editor-in-Chief

PLOS Pathogens

orcid.org/0000-0002-7699-2064